# Clinical Outcomes of Everolimus Rechallenge in Patients with Pancreatic Neuroendocrine Neoplasms with No Other Treatment Options

**DOI:** 10.3390/cancers14225669

**Published:** 2022-11-18

**Authors:** Yusuke Kurita, Noritoshi Kobayashi, Kazuo Hara, Nobumasa Mizuno, Takamichi Kuwahara, Nozomi Okuno, Shin Haba, Motohiko Tokuhisa, Sho Hasegawa, Kensuke Kubota, Atsushi Nakajima, Yasushi Ichikawa

**Affiliations:** 1Department of Gastroenterology and Hepatology, Yokohama City University Hospital, Yokohama 236-0004, Japan; 2Department of Gastroenterology, Aichi Cancer Center Hospital, Nagoya 464-8681, Japan; 3Department of Oncology, Yokohama City University Hospital, Yokohama 236-0004, Japan

**Keywords:** pancreas, neuroendocrine neoplasms, everolimus

## Abstract

**Simple Summary:**

Pancreatic neuroendocrine neoplasms (PNENs) are a type of pancreatic tumor. Everolimus is an oral inhibitor of the mammalian target of rapamycin (mTOR). Although PNENs respond well to systematic treatment, including everolimus, the treatment options are often exhausted in clinical practice with the progression of the tumor. Recently, the efficacy of molecular targeted drugs, including mTOR inhibitors, has been assessed in rechallenge experiments. The efficacy and safety of rechallenge with everolimus in PNENs remains unclear. This study retrospectively evaluated the treatment outcomes of patients who received everolimus rechallenge. Everolimus rechallenge may be a drug treatment option for those with advanced PNEN for whom no other drug treatment option is available.

**Abstract:**

Background: The clinical outcomes of everolimus rechallenge in patients with pancreatic neuroendocrine neoplasms (PNENs) are unknown. This study aimed to investigate the treatment outcomes and safety of everolimus rechallenge treatment with PNENs. Methods: Clinical data of everolimus-treated patients with PNENs at two institutions were collected. Patients who underwent everolimus rechallenge were included in the study. We analyzed the progression-free survival (PFS) and treatment response associated with everolimus rechallenge and the adverse events. Results: Between 2008 and 2020, 117 patients received initial treatment with everolimus, of which 14 patients received everolimus rechallenge. With regard to the grade of PNENs, there were 2 cases of G1, 11 cases of G2, and 1 case of G3. The median rechallenge PFS was 5.7 months. The objective response rate was 21.4%. the disease control rate was 71.4%. The only major grade 3 or 4 adverse event was neutropenia (*n* = 1, 7.1%). No other severe adverse event was observed. Conclusion: The outcomes and safety of everolimus rechallenge were verified, and it was deemed an acceptable treatment. Everolimus rechallenge may provide a new drug therapy for patients with advanced PNENs for whom no other drug treatment option is available.

## 1. Introduction

Pancreatic neuroendocrine neoplasms (PNENs) are a type of pancreatic tumors [1]. Everolimus is an oral inhibitor of the mammalian target of rapamycin (mTOR). It has shown efficacy in the management of PNENs in RADIANT-3 [2]. In addition to everolimus, PNENs respond well to systematic treatment [3]. A variety of therapies are considered useful, including surgery [4,5], sunitinib [6], streptozocin [7,8,9], and peptide receptor radionuclide therapy [10]. However, in clinical practice, the treatment options are often exhausted as the tumor progresses.

Recently, the efficacy of rechallenges with molecular targeted drugs [11,12] and cytotoxic agents [13,14,15] has been demonstrated in colorectal cancer chemotherapy. Moreover, a previous study reported the effectiveness of rechallenges with mTOR inhibitors in patients with metastatic renal cell carcinoma [16]. This report examined firstly received everolimus or temsirolimus and other secondarily received mTOR inhibitors. In this study, 6 of 12 patients (50%) responded to rechallenge with everolimus, and the median treatment durations for the everolimus → temsirolimus and temsirolimus → everolimus sequences were 10.3 months and 5.8 months, respectively. However, there are no previous reports about everolimus rechallenge therapy for patients with advanced PNENs. 

We recently initiated everolimus rechallenge therapy for patients with advanced PNENs who had no other treatment options. The therapy was initiated only at the patient’s request. However, the efficacy and safety of everolimus in PNENs remain unclear. The study evaluated the treatment outcomes of the patients who received everolimus rechallenge retrospectively.

## 2. Methods

The clinical data of the patients were retrospectively collected from the Yokohama City University Hospital and Aichi Cancer Center Hospital between 2008 and 2020. The inclusion criteria were as follows: patients with histologically proven advanced PNENs based on biopsy or surgical specimen findings, and patients treated with everolimus rechallenge. This study was approved by the institutional review boards of Yokohama City University Hospital (B200500063) and Aichi Cancer Center Hospital (20201326). Only medical information was used, and there was no invasion of the participants’ privacy in this observational study. All the patients received an opt-out form for informed consent.

The patients’ characteristics were evaluated within the 4 weeks before everolimus rechallenge initiation. Clinicopathological data were collected, including the patient’s age, sex, hereditary status, functional or nonfunctional tumor, metastatic site, and treatment history, including the initial introduction of everolimus. The tumors were evaluated using contrast computed tomography (CT) before everolimus rechallenge administration. 

The pathological grading was reviewed by a specialist using information obtained from pathological reports. The Ki-67 proliferation index was calculated, and the tumor grade was determined according to the 2019 WHO classification. 

### 2.1. Treatment of Everolimus Rechallenge

In this study, everolimus rechallenge was defined as a case in which another treatment was introduced for at least three months or there was a treatment-free follow-up period of at least six months after the initial application of everolimus was terminated for any reason. Everolimus rechallenge was performed only when there was no other treatment option and the patient agreed to and requested the intervention. 

Everolimus treatment was started at a daily dose of 10 mg, and the initial dose was reduced to 5 mg depending on the clinical status of the patient. The treatment response was evaluated according to the Response Evaluation Criteria in Solid Tumors (RECIST, version 1.1). For the evaluation after the everolimus treatment, CT scans were performed every 3 months after the start of the treatment to re-evaluate the disease. Everolimus was discontinued when there was disease progression or unacceptable toxicity, based on the patient’s preference, and an alternative treatment, such as surgery, was feasible. The doses were delayed or reduced according to the physician’s decision if there were adverse events. Adverse events were investigated based on the Common Terminology Criteria for Adverse Events, version 4.0. A dose reduction of everolimus was required for one or more of the following events: febrile neutropenia, grade 3 or 4 neutropenia, grade 3 or 4 thrombocytopenia, any other grade 3 or 4 toxicity, and a delayed recovery from toxicity by more than 2 weeks. Treatment was discontinued if interstitial pneumonitis of grades 2, 3, or 4 developed. 

### 2.2. Endpoints

The primary endpoint of this study was progression-free survival (PFS) associated with everolimus rechallenge. The secondary endpoints were treatment response to everolimus rechallenge and adverse events due to everolimus rechallenge.

### 2.3. Statistical Analyses

PFS was defined as the period from everolimus initiation to disease progression or death from any cause. The PFS was estimated using the Kaplan–Meier method, and differences between curves were evaluated using the log-rank test. For the analysis of the associations between factors, the Mann–Whitney U test was used for the continuous variables and Pearson’s χ^2^ or Fisher’s exact test was used for the categorical data, and *p* values < 0.05 were considered significant. Statistical analyses were performed using the SPSS version 28 software (IBM, Armonk, NY, USA). 

## 3. Results 

### 3.1. Patients

Between 2008 and 2020, 117 patients received initial treatment with everolimus, of which 14 patients received everolimus rechallenge. Demographic and other characteristics of the patients upon everolimus initiation are summarized in Table 1. The median patient age was 53 (range, 32–70) years, and five of the patients were male (35.7%) and nine were female (64.3%). The WHO grades (2019) were G1, G2, and G3 in 2 patients (14.3%), 11 patients (78.6%), and 1 (7.1%) patient, respectively. The objective response rate for the initial introduction of everolimus was 28.6%. The control rate for the initial introduction of everolimus was 78.6%, and the median PFS for the initial introduction of everolimus was 8.7 (95% CI, 0.9–16.4) months.

### 3.2. Progression-Free Surival and Response Rate for Everolimus Rechallenge

The rates of PFS with everolimus rechallenge are shown in Figure 1. The median PFS with everolimus rechallenge was 5.7 (95% CI, 1.0–10.5) months. Of the 14 patients who received the reintroduction of everolimus, 0 had a complete response (CR), 3 had a partial response (PR), 7 had stable disease (SD), and 4 had progressive disease (PD) (Table 2). The objective response rate was 21.4%. The disease control rate was 71.4%.

Figure 2 shows a swimmer plot of the progress following the initial introduction of everolimus, subsequent other treatments, and everolimus rechallenge.

Figure 3 shows a case in which everolimus rechallenge produced a PR effect. This case in Figure 3 is case 7 in Figure 2. This patient was treated with everolimus, and PD was terminated at the time of initial introduction of everolimus. The PFS after the initial introduction of everolimus was 13.7 months. After other treatments, everolimus rechallenge was performed. The everolimus rechallenge showed a 35% reduction in the target tumor size from 34 mm (before everolimus rechallenge) to 22 mm 3 months after the start of the everolimus rechallenge, confirming a favorable effect.

### 3.3. Adverse Events due to Everolimus Rechallenge

Table 3 summarizes the adverse events due to the treatment of rechallenge and the initial introduction of everolimus. In the context of everolimus rechallenge, the only major grade 3 or 4 adverse event was neutropenia (*n* = 1, 7.1%). One grade 3 adverse event that included neutropenia, rash, and interstitial pneumonia occurred during the initial introduction of everolimus. However, none of the patients who experienced a grade 3 adverse event at the time of the initial introduction developed a grade 3 or higher adverse event at the time of rechallenge.

For the initial introduction of everolimus, the all-grade adverse events were anemia (*n* = 1, 7.1%), thrombocytopenia (*n* = 3, 21.4%), leukopenia (*n* = 1, 7.1%), neutropenia (*n* = 3, 21.4%). The non-hematologic all-grade toxicities were stomatitis (*n* = 7, 50.0%), rash (*n* = 4, 28.6%), decreased appetite (*n* = 3, 21.4%), noninfectious pneumonitis (*n* = 1, 7.1%), and hyperglycemia (*n* = 1, 7.1%). Other severe adverse events were not observed.

## 4. Discussion

In this study, we investigated the clinical outcomes and safety of rechallenge with everolimus, an mTOR inhibitor, in patients with advanced PNENs. The mTOR pathway is a central regulator of cellular proliferation. Mutations associated with the mTOR pathway have been detected in 18% of neuroendocrine tumors [17]. mTOR is the core component of two complexes, mTORC1 and mTORC2, which have distinct signaling pathways and functions [18]. Everolimus may only partially inhibit mTORC1 and incompletely block its given downstream target, especially 4E-BP1, according to some studies [19,20]. Additionally, the inability of everolimus to block mTORC2 induces upstream Akt phosphorylation [21]. Emerging evidence shows that the dual inhibition of these pathways, PI3K/Akt and the mTOR pathway, could be a novel therapeutic target used to overcome everolimus resistance. The efficacy and safety of the dual PI3K/mTOR inhibitor, BEZ235, was examined in patients with advanced NETs [22]. However, a high toxicity of BEZ235 was observed and led patients to require frequent dose modifications and treatment discontinuations in this study. Of course, other mechanisms of resistance to mTOR inhibitors have also been demonstrated, including the activation of mitogen-activated protein kinase (MAPK); upregulation of pro-angiogenic factors; and activation of the Ras pathway [21,23,24]. Unfortunately, no clinical trials have examined everolimus resistance. Previous clinical data on rechallenges with mTOR inhibitors were used to examine firstly received everolimus or temsirolimus and other secondarily received mTOR inhibitors in patients with metastatic renal cell carcinoma [16]. In this study, 6 of 12 patients (50%) responded to rechallenge with everolimus, and the median treatment durations for the everolimus → temsirolimus and temsirolimus → everolimus sequences were 10.3 months and 5.8 months.

In this study, there were six patients who had terminated everolimus treatment due to PD at the time of initial introduction of everolimus, of whom had PD, two had SD, and one had PR as the treatment response to rechallenge with everolimus. In short, half of the patients with PD whose treatment was terminated at initial introduction of everolimus were able to achieve disease control at the time of everolimus rechallenge. Therefore, residual tumor cells that are sensitive to everolimus may be alive at initial introduction and grow for a long duration with other treatment options, and the tumor mass may be shrunk, and the patient may achieve stable disease and a partial response to everolimus rechallenge. In this regard, since this study was a retrospective analysis of a small number of cases, a detailed analysis that includes insights into the molecular mechanisms is required in future. Furthermore, studies involving higher number of cases are desirable so as to derive definitive conclusions on the topic.

Compared to the previously reported findings on the initial introduction of everolimus (median PFS was 11 months) [2], the median PFS for the everolimus rechallenge in this study was shorter (median PFS was 5.7 months). However, the objective response rate and disease control rate were 21.4 and 71.4% in this study. The objective response rate and disease control rate were 4.8 and 77.8% for Radiant-3 [2] and 19.0 and 85.7% according to the previous reports in Japan [25], which are close to the results of this study. There were some cases of PD with everolimus rechallenge. However, there were cases in which everolimus was effective for more than one year, even in patients whose treatment was terminated due to the progression of the disease or adverse events after the initial introduction of everolimus. One case of each grade 3 adverse event (neutropenia, rash, and interstitial pneumonia) occurred upon the initial introduction of everolimus, but none of the patients who experienced grade 3 adverse events developed grade 3 or higher serious adverse events upon rechallenge. There were few serious adverse events in the patients treated with everolimus rechallenge. Consequently, the rechallenge was considered safe. Everolimus rechallenge may provide a new treatment option for patients with advanced PNENs for whom no other treatment option is available. 

This study has several limitations. Firstly, it was retrospective in nature. Secondly, this study included a small number of patients on whom everolimus rechallenge was performed. In two of the three PR cases that involved rechallenges, the factor that terminated the initial introduction was not the PD decision. It is difficult to discuss the efficacy of everolimus rechallenge. On the other hand, there were cases in which a relatively long-term SD decision was obtained, even in patients with a PD decision at the time of the initial introduction, and there were cases in which treatment was possible without serious adverse events, even in cases in which the initial introduction was terminated due to adverse events. Although we believe that the effect on tumor shrinkage is insufficient, we believe that this report is meaningful, considering the relatively mild characteristics of molecular targeted drugs. Larger prospective clinical trials of everolimus rechallenge are necessary to confirm the results of the present study.

## 5. Conclusions

In conclusion, the outcomes and safety of everolimus rechallenge were verified, and it was deemed an acceptable treatment. Everolimus rechallenge may provide a new drug therapy for patients with advanced PNENs for whom no other drug treatment option is available.

## Figures and Tables

**Figure 1 cancers-14-05669-f001:**
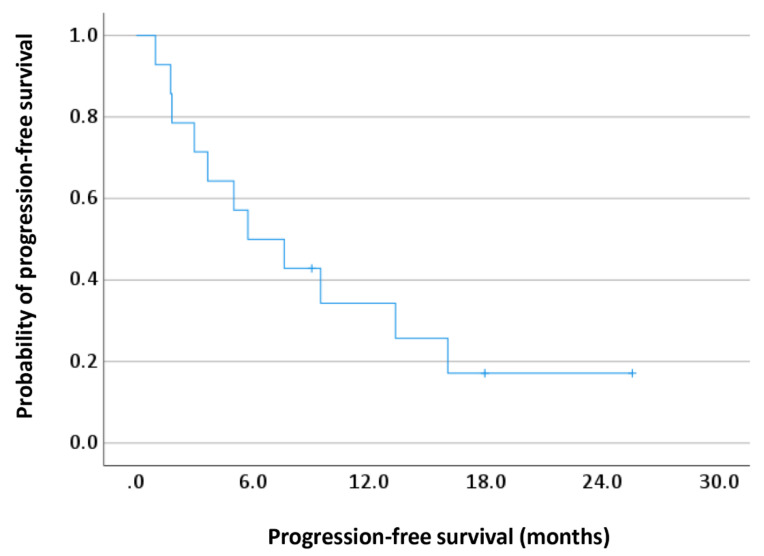
Progression-free survival with everolimus rechallenge. Kaplan–Meier curves for progression-free survival (PFS) with everolimus rechallenge. The median PFS with everolimus rechallenge was 5.7 (95% CI, 1.0–10.5) months.

**Figure 2 cancers-14-05669-f002:**
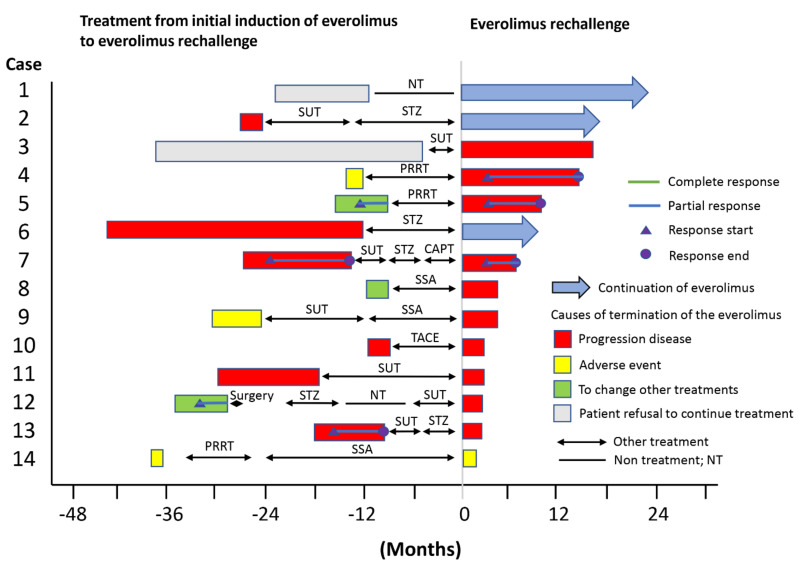
Swimmer plot data for everolimus rechallenge. Swimmer plot showing everolimus rechallenge and the initial introduction of everolimus and other treatments before rechallenge. CAPT, capecitabine and temozolomide; NT, non-treatment; PRRT, peptide receptor radionuclide therapy; SSA, sandostatin analogue; STZ, streptozocin; SUT, *sunitinib*; TACE, transcatheter arterial chemoembolization.

**Figure 3 cancers-14-05669-f003:**
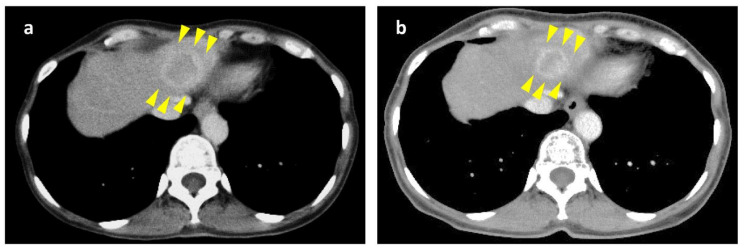
Cases with a good response to rechallenge with everolimus. (**a**) Contrast CT before everolimus rechallenge; target tumor diameter 34 mm. (**b**) Contrast CT 3 months after everolimus rechallenge; target tumor diameter 22 mm; PR response achieved.

**Table 1 cancers-14-05669-t001:** Patients’ characteristics with everolimus rechallenge.

	Rechallenge (*n* = 14)
Age, median (range), year	53 (32–70)
Sex (%) male/female	5 (35.7)/9 (64.3)
WHO grade (2019) (%)	
G1 G2 G3	2 (14.3) 11 (78.6) 1 (7.1)
Ki-67 index, median (range) *	5.0 (2.0–41.8)
Hereditary status (%)	
Sporadic/familial	100 (100)/0
Functionality (%)	
Function/non-function	1 (7.1)/13 (92.9)
Metastasis (%)	
Lymph node/liver	2 (14.3)/14 (100)
Performance status (%)	
0/1/2/3–4	12 (85.7)/1 (7.1)/1 (7.1)/0
Treatment line of the initial introduction of everolimus, mean (range)	2.1 (1–5)
Tumor response to the initial introduction of everolimus (%)	
CR PR SD PD	0 4 (28.6) 7 (50.0) 3 (21.4)
Objective response rate for the initial introduction of everolimus (%)	4 (28.6)
Disease control rate for the initial introduction of everolimus (%)	11 (78.6)
The median PFS for the initial introduction of everolimus (95% CI), month	8.7 (0.9–16.4)
Causes of discontinuation of the initial introduction of everolimus (%)	
Progression disease Adverse event Change to other treatments Patient refusal to continue treatment	6 (42.9) 3 (21.4) 2 (14.3) 2 (14.3)
Treatment line of everolimus rechallenge, mean (range)	4.5 (2–7)
Everolimus rechallenge plus SSA (%)	7 (50.0)
Time from the initial everolimus to everolimus rechallenge treatment	
Median, months (range) ≤6 mo >6 mo to ≤2 y >2 y to ≤5 y >5 y	12.7 (4.0–36.0) 1 (7.1) 9 (64.3) 3 (21.4) 0
Time from the initial diagnosis to everolimus rechallenge treatment	
Median, months (range) ≤6 mo >6 mo to ≤2 y >2 y to ≤5 y >5 y	44.5 (11.6–193.1) 0 1 (7.1) 9 (64.3) 4 (28.6)

* Three patients with an unknown Ki-67 index were excluded. MEN, multiple endocrine neoplasia; SSA, somatostatin analogues; WHO, World Health Organization.

**Table 2 cancers-14-05669-t002:** Tumor response and progression-free survival with everolimus rechallenge.

	Overall (*n* = 14)	G1 (*n* = 2)	G2 (*n* = 11)	G3 (*n* = 1)
Tumor response (%)				
CR PR SD PD	0 3 (21.4) 7 (50.0) 4 (28.6)	0 0 2 (100) 0	0 3 (27.3) 4 (36.4) 4 (36.4)	0 0 1 (100) 0
Objective response rate (%)	3 (21.4)	0	3 (27.3)	0
Disease control rate (%)	10 (71.4)	2 (100)	7 (63.6)	1 (100)
Median PFS (95% CI), month	5.7 (1.0–10.5)	16.0 (-)	5.0 (0.0–10.0)	5.8 (-)

Objective response rate = CR + PR. Disease control rate = CR + PR + SD. CR, complete response; PFS, progression-free survival; PD, progressive disease; PR partial response, SD stable disease.

**Table 3 cancers-14-05669-t003:** Toxicity of everolimus during rechallenge and at the time of the initial introduction (*n* = 14).

	Initial Introduction		Rechallenge	
	All Grades	Grade 3/4	All grades	Grade 3/4
Hematological toxicity				
Anemia	3 (21.4)	0	1 (7.1)	0
Thrombocytopenia	0	0	3 (21.4)	0
Leukopenia	1 (7.1)	0	1 (7.1)	0
Neutropenia	2 (14.3)	1 (7.1)	3 (21.4)	1 (7.1)
Febrile neutropenia	0	0	0	0
Non-hematological toxicity				
Stomatitis *	8 (57.1)	0	7 (50.0)	0
Rash	3 (21.4)	1 (7.1)	4 (28.6)	0
Decreased appetite	1 (7.1)	0	3 (21.4)	0
Headache	0	0	1 (7.1)	0
Noninfectious pneumonitis ‡	4 (28.6)	1 (7.1)	1 (7.1)	0
Hyperglycemia	1 (7.1)	0	1 (7.1)	0

* Included in this category are stomatitis, aphthous stomatitis, mouth ulceration, and tongue ulceration. ‡ Included in this category are pneumonitis, interstitial lung disease, lung infiltration, and pulmonary fibrosis.

## Data Availability

Data are available on request because of restrictions, e.g., privacy or ethics.

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
