# Peer review of "Clinical Outcomes of Everolimus Rechallenge in Patients with Pancreatic Neuroendocrine Neoplasms with No Other Treatment Options"

_cancers, 2022, doi:10.3390/cancers14225669_

Round 1

Reviewer 1 Report

I have checked the manuscript (title: Feasibility of everolimus rechallenge in patients with pancreatic neuroendocrine neoplasms with no other treatment options). I enumerate some comments as follows.  

Major points

1.     Therapeutic effects of initial induction of everolimus are lacking in both the text and Table 1. Authors should add them.

2.     There are only 6 cases in which everolimus was discontinued by progressive disease. Threrefore, it is postulated that the sensitivity was restored irrespective of resistant to mTOR inhibitors in many cases treated with everolimus rechallenges. Authors should add the description in this manuscript.

Minor point

1.     Authors should indicate therapeutic effects of not only complete or partial response but also stable disease in both initial induction and rechallenge of everolimus in Figure 2.

2.     If possible, please show the figures in the cases in which everolimus rechallenges were effective.

Author Response

Major points

  1. Therapeutic effects of initial induction of everolimusare lacking in both the text and Table 1. Authors should add them.

Response: We thank the reviewer for the comment. Accordingly, we have included the therapeutic effects of initial introduction of everolimus in the text (line 122-124) and Table 1.

  1. There are only 6 cases in which everolimus was discontinued by progressive disease. Threrefore, it is postulated that the sensitivity was restored irrespective of resistant to mTOR inhibitors in many cases treated with everolimus rechallenges. Authors should add the description in this manuscript.

Response: We thank the reviewer for the insightful comment. We concur with the views of the reviewer. Accordingly, we have incorporated a note in the Discussion section regarding the possibility that in many cases in which everolimus was re-administered, sensitivity to the drug may have been restored, regardless of resistance to mTOR inhibitors. (line 205-208)

Minor point

  1. Authors should indicate therapeutic effects of not only complete or partial response but also stable disease in both initial induction and rechallenge of everolimus in Figure 2.

Response: We thank the reviewer for valuable input. We have added additional information on stable diseases in Figure 2.

  1. If possible, please show the figures in the cases in which everolimus rechallenges were effective.

Response: We thank the reviewer for the informative comments. Accordingly, we have added a case of PR, as shown in Figure 3. In Figure 3, we depict a case wherein  everolimus challenge produced a PR effect. This is illustrated in Case 7 of Figure 2. The patient was treated with everolimus challenge and PD was terminated at the time of initial everolimus induction. The PFS after the initial everolimus administration was 13.7 months. After other treatments, an everolimus rechallenge was performed. Everolimus rechallenge resulted in 35% reduction in target tumor size from 34 mm (before everolimus rechallenge) to 22 mm, 3 months after the start of the everolimus rechallenge. This confirmed a favorable effect of everolimus.

Reviewer 2 Report

The authors report the clinical effects of everolimus rechallenge in pancreatic neuroendocrine tumors.

My comments are as follows; 

1. Unfortunately, it is impossible to assess the efficacy of everolimus rechallenge because of small size study.

2. Please discuss the safety and risk of everolimus rechallenge in patients who have experience of adverse events during initial administration of everolimus.

3. In the last line of page 4, the sentence might be incorrect. The authors should add “and” between “after everolimus rechallenge” and “before everolimus rechallenge”.

Author Response

  1. Unfortunately, it is impossible to assess the efficacy of everolimus rechallenge because of small size study.

Response: Thank you for your comment. As you pointed out, we believe it is difficult to assess efficacy, and we have stated that it is difficult to discuss efficacy in the limitations as well (line 228-232). We have removed the sentence stating that everolimus rechallenge is effective from the text and revised the wording to say that everolimus may be feasible in the future.

  1. Please discuss the safety and risk of everolimus rechallenge in patients who have experience of adverse events during initial administration of everolimus.

Response: Thank you for your comment. There were three cases in which everolimus was terminated because of adverse events at the initial induction. One case each of grade 3 adverse events (neutropenia, rash, and interstitial pneumonia) occurred at the initial induction of everolimus, but none of the patients who experienced grade 3 adverse events developed grade 3 or higher serious adverse events at rechallenge. Even if a grade 3 or higher adverse event occurs at the initial induction, no serious adverse event occurs at rechallenge, and rechallenge may be administered relatively safely.

The above has been added to the Results (line 171-175) and Discussion (line 220-223) section of the main text. We also included additional information on adverse events at the initial induction (Table 3).

  1. In the last line of page 4, the sentence might be incorrect. The authors should add “and” between “after everolimus rechallenge” and “before everolimus rechallenge”.

Response: Thank you for your important comments. We have made the abovementioned corrections.

Reviewer 3 Report

The Authors present a study evaluating the feasibility of Everolimus rechallenge in patients pretreated for metastatic neuroendocrine tumors of the pancreas.

Rechallenged patients were retrospectively selected from charts of two different institutions.

According to Authors, presented data indicate that the treatment is well tolerated and endowed with some activity.

Although it is well known that retrospective feasibility studies come with unavoidable bias, they could be justified in instances of rare diseases, such as pancreatic neuroendocrine tumor, when prospective studies are difficult to be considered.

In any case a few issues need to be address before considering the paper for publication.

1)    Background for the study is not sufficiently provided. What is the rationale for the mTOR re-challenge? Is there preclinical evidence of molecular mechanisms supporting effectiveness of mTOR rechallenge? In the cited clinical study (ref 16), mTOR re-challenge was done with everolimus and temsirolimus. Are there any clinical studies employing exclusively everolimus for re-challenge? 

2)    The wide period of time considered when retrieving cases (more than 10 years) may considerably affect several aspects of patient management spanning from side effects management to response evaluation. Although some of them can’t be adequately accounted for, given the retrospective nature of the study, frequency of response evaluation, which may have been different in different patients, would be critical in the determination of PFS. Authors should clearly provide disease re-evaluation policies of the two Institution involved in the study

3)    Does the reported burden of liver metastasis refer to the extent of liver volume substituted by secondary lesions? If so please clearly state this.

4)    In order to evaluate salvage therapy for heavily pretreated cancer, of foremost importance is patient Performance Status. Notably, no information as been provided about PS of patients rechallenged with everolimus.

5)    The observed response and disease control rates should be considered with extreme caution as they might represent random consequences of the extremely reduced size of the study and the highly heterogeneous population. In fact, if one carefully considers the treatment history of these patients as depicted in the provided swimmer plot, 2/3 of the PR were observed in patients for which the initial treatment with everolimus was prematurely interrupted either for toxicity (case 4) or for patients’ refusal, then in the absence of a real progression. These two may represent cases in which tumors never became resistant to everolimus and therefore, they were not really “rechallenged”.

6)    Authors should more clearly discuss and highlight limitations of their reported data, and tune down statements at lines 153-163 regarding safety and efficacy of everolimus re-challenge in PNEN.

7)    The manuscript needs a careful English revision, since there are typos also in the abstract (e.g.: lines 24-25, what is the meaning of “Despite the limited number of patients, Everolimus re-challenge may be effective with safe” ?

Author Response

1) Background for the study is not sufficiently provided. What is the rationale for the mTOR re-challenge? Is there preclinical evidence of molecular mechanisms supporting effectiveness of mTOR rechallenge? In the cited clinical study (ref 16), mTOR re-challenge was done with everolimus and temsirolimus. Are there any clinical studies employing exclusively everolimus for re-challenge?

Response: Thank you for your valuable comments. mTOR is the core components of 2 complexes, mTORC1 and mTORC2, which have distinct signaling pathways and functions. Everolimus specifically directly inhibits the mTORC1 and its downstream p70S6K/4E-BP1, resulting in attenuation of protein synthesis as well as cell growth. Several studies showed that everolimus may only partially inhibit mTORC1 and incompletely block its certain downstream target, especially 4E-BP1. The inability of everolimus to block mTORC2 induces upstream Akt phosphorylation. Emerging evidence shows that dual inhibition of these pathways, particularly PI3K/Akt and the mTOR pathway, could be a novel therapeutic approach to overcome everolimus resistance. The efficacy and safety of the dual PI3K/mTOR inhibitor, BEZ235 was examined in patients with advanced NETs. However, high toxicity of BEZ235 observed in this study led patients to require frequent dose modifications and treatment discontinuations. Unfortunately, there is no other clinical study to overcome everolimus resistance. We described these preclinical evidence and clinical trial in discussion section. (line 177-193)

Previous reports of mTOR inhibitor rechallenge have only examined firstly received everolimus or temsirolimus and secondary received other mTOR inhibitor. In this literature, 6 of 12 patients (50%) responded to rechallenge everolimus and median treatment duration for everolimus → temsirolimus and temsirolimus → everolimus sequences were 10.3 months and 5.8 months, respectively. In this study, author concluded that despite the limited number of patients, this highlights the feasibility of utilizing mTOR rechallenge as an integral part of sequential treatment strategies in metastatic renal cell carcinoma.

We have now reported on the everolimus challenge. There have been no reports of rechallenge with everolimus alone, which we believe is the novelty of this study. We have modified the introduction slightly (line 50-54).

2)    The wide period of time considered when retrieving cases (more than 10 years) may considerably affect several aspects of patient management spanning from side effects management to response evaluation. Although some of them can’t be adequately accounted for, given the retrospective nature of the study, frequency of response evaluation, which may have been different in different patients, would be critical in the determination of PFS. Authors should clearly provide disease re-evaluation policies of the two Institution involved in the study

Response: Thank you for the important suggestion. For evaluation after everolimus treatment was started, CT scans were taken every 3 months after the start of treatment to reevaluate the disease (line 93-94).

3)    Does the reported burden of liver metastasis refer to the extent of liver volume substituted by secondary lesions? If so please clearly state this.

 Response) Thank you for your comment. The liver metastasis burden represents the metastatic tumor volume in the liver parenchyma. The liver metastasis burden was classified as <10%, 10-25%, 25-50%, and >50% based on CT findings within 4 weeks of everolimus initiation. We have added this note to the text (line 75-80).

4)    In order to evaluate salvage therapy for heavily pretreated cancer, of foremost importance is patient Performance Status. Notably, no information as been provided about PS of patients rechallenged with everolimus.

 Response: Thank you for your comment. The performance statuses are listed in Table 1.

5)    The observed response and disease control rates should be considered with extreme caution as they might represent random consequences of the extremely reduced size of the study and the highly heterogeneous population. In fact, if one carefully considers the treatment history of these patients as depicted in the provided swimmer plot, 2/3 of the PR were observed in patients for which the initial treatment with everolimus was prematurely interrupted either for toxicity (case 4) or for patients’ refusal, then in the absence of a real progression. These two may represent cases in which tumors never became resistant to everolimus and therefore, they were not really “rechallenged”.

Response: Thank you for your comment. Indeed, as you pointed out, in two of the three PR cases of rechallenge, the factor that terminated the initial induction was not the PD decision. This point should be carefully considered, and we consider it a limitation. On the other hand, there were cases in which a relatively long-term SD decision was obtained, even in patients with a PD decision at the time of initial induction, and there were cases in which treatment was possible without serious adverse events, even in cases in which initial induction was terminated due to adverse events. Although we believe that the effect on tumor shrinkage is insufficient, we believe that this report is meaningful considering the relatively mild characteristics of molecular-targeted drugs. Based on your comments, we have added limitation and corrected the Discussion (line 228-239).

6)     Authors should more clearly discuss and highlight limitations of their reported data, and tune down statements at lines 153-163 regarding safety and efficacy of everolimus re-challenge in PNEN.

Response: Thank you for your suggestion. Based on the limitations of this data, we have toned down our discussion and revised it as follows:

“In this study, we investigated the feasibility and safety of rechallenge with everolimus, an mTOR inhibitor, in patients with advanced PNENs. The mTOR pathway is a central regulator of cellular proliferation. Mutations associated with the mTOR path-way have been detected in 18% of neuroendocrine tumors [17]. mTOR is the core com-ponents of 2 complexes, mTORC1 and mTORC2, which have distinct signaling pathways and functions [18]. Everolimus may only partially inhibit mTORC1 and incompletely block its certain downstream target, especially 4E-BP1 by some studies [19,20]. And also the inability of everolimus to block mTORC2 induces upstream Akt phosphorylation [21]. Emerging evidence shows that dual inhibition of these pathways, PI3K/Akt and the mTOR pathway, should be a novel therapeutic target to overcome everolimus resistance. The efficacy and safety of the dual PI3K/mTOR inhibitor, BEZ235 was examined in pa-tients with advanced NETs [22]. However, high toxicity of BEZ235 observed and led pa-tients to require frequent dose modifications and treatment discontinuations in this study. Of course, other mechanisms of resistance to mTOR inhibitors have also demonstrated including; activation of mitogen activated protein kinase (MAPK); up-regulation of pro-angiogenic factors; and activation of Ras pathway [21,23,24]. Unfortunately, there is no other clinical trials to overcome everolimus resistance.  In previous clinical data of rechallenges with mTOR inhibitors, firstly received everolimus or temsirolimus and secondary received other mTOR inhibitor for the patients with metastatic renal cell car-cinoma [16]. In this literature, 6 of 12 patients (50%) responded to rechallenge everolimus and median treatment duration for everolimus → temsirolimus and temsirolimus → everolimus sequences were 10.3 months and 5.8 months.

In this study, there were six cases that were terminated with PD at the time of initial induction of everolimus, and three of them had PD as the treatment response to rechallenge with everolimus. Two of the cases had SD, while one had PR. Therefore, in some cases, further therapy without mTOR inhibitors could restore the sensitivity to mTOR inhibitors in tumor cells that are resistant to mTOR inhibitors. Therefore, re-challenge with mTOR inhibitors may result in the recovery of antitumor effects in the later stages of treatment. However, in this study, there were only six cases in which everolimus administration was discontinued due to disease progression. We speculate that in some cases where everolimus was re-administered, the sensitivity of everolimus may have been restored, regardless of resistance to the mTOR inhibitor. In this regard, since this study was a retrospective analysis of a small number of cases, a detailed analysis that included insights on molecular mechanisms is required in future. Further, studies involving higher number of cases are desirable to derive definitive conclusions on the topic.” (Line 177-211)

7)    The manuscript needs a careful English revision, since there are typos also in the abstract (e.g.: lines 24-25, what is the meaning of “Despite the limited number of patients, Everolimus re-challenge may be effective with safe” ?

Response: Thank you for this comment. We have corrected the text in the areas mentioned above.

Conclusion were as follows. 

In conclusion, everolimus rechallenge may be feasibile and safety treatment. everolimus rechallenge may be a new treatment option for patients with advanced PNENs for whom no other treatment option is available.

Round 2

Reviewer 1 Report

I have re-checked the manuscript (title: Feasibility of everolimus rechallenge in patients with pancreatic neuroendocrine neoplasms with no other treatment options). I enumerate some comments as follows.

Minor point

1.     The sentence ‘Two of the cases had SD, while one had PR.’ should be linked to former sentence in the page 8, line 309 in Discussion section.

2.     The sentence ‘However, in this study, there were only six cases in which everolimus administration was discontinued due to disease progression.’ should be deleted and the word from ‘some’ to ‘another’ changed in the page 8, line 313-314 in Discussion section.

Author Response

Minor point

  1. The sentence ‘Two of the cases had SD, while one had PR.’ should be linked to former sentence in the page 8, line 309in Discussion section.

Response) Thank you for your very valuable comments. We have revised the text as follows

“In this study, there were six cases that were terminated with PD at the time of initial introduction of everolimus, of which 3 had PD, 2 had SD, and 1 had PR as the treatment response to rechallenge with everolimus.”

  1. The sentence ‘However, in this study, there were only six cases in which everolimus administration was discontinued due to disease progression.’ should be deleted and the word from ‘some’ to ‘another’ changed in the page 8, line 313-314in Discussion section.

Response) Thank you for your very valuable comments. We have corrected the text. Subsequently, the text was partially deleted and revised based on the reviewer 3's opinion as well.

Reviewer 2 Report

The manuscript is properly revised according to the reviewers' comments. 

Author Response

Thank you very much for your peer review.
Best regards.

Reviewer 3 Report

The manuscript was improved. However, the following issues remain to be more accurately addressed.

1)    The provided description of the “burden of liver metastasis” is not sufficiently clear. Saying that tumor burden “represents the metastatic volume in the liver parenchyma” does not immediately justify the way it was quantified, i.e. as a percentage

2)    As noted in my previous comment, it is not possible to state that rechallenge restores sensitivity to everolimus, since in 2 of the 3 observed PR with rechallenge, the reason for initial discontinuation of the drug was not progression, but patient refusal or toxicity. Therefore, it is not possible to assume that in these cases resistance to the drug developed.

I think that the sentence added by Authors (“ We speculate that in some cases where everolimus was re-administered, the sensitivity of everolimus may have been restored, regardless of resistance to the mTOR inhibitor”) appears unclear and does not appear to adequately account for my original criticism. In other word, it would seem quite a nonsense to state that “rechallenge restore sensitivity to everolimus even in the absence of resistance”, if I have not misinterpreted Authors’ words.

3)    English needs major revision!   It appears that the Authors just reviewed their text, which in several sentences, scattered throughout the paper.

Few examples of this are:

a)    Everolimus rechallenge may be feasibile and safety treatment. Everolimus rechallenge may be a new treatment option for patients with advanced PNENs for whom no other treatment option is available”.

b)    Few treatment options were available for unresectable PNENs. Therefore, physicians sometime must recommend feasible treatment for the advanced PNENs patients with terminated all available treatment and no other treatment options in high volume center.

The sentence is hardly understandable  

c)    Unfortunately, there is no other clinical trials to overcome everolimus resistance.

Clinical trials do not overcome drug resistance!

Author Response

The manuscript was improved. However, the following issues remain to be more accurately addressed.

  • The provided description of the “burden of liver metastasis” is not sufficiently clear. Saying that tumor burden “represents the metastatic volume in the liver parenchyma” does not immediately justify the way it was quantified, i.e. as a percentage

Response)Thank you for your valuable comments. As you indicated, burden of liver metastasis is difficult to be objective and justified, so we removed it from this study.

2)    As noted in my previous comment, it is not possible to state that rechallenge restores sensitivity to everolimus, since in 2 of the 3 observed PR with rechallenge, the reason for initial discontinuation of the drug was not progression, but patient refusal or toxicity. Therefore, it is not possible to assume that in these cases resistance to the drug developed.

I think that the sentence added by Authors (“ We speculate that in some cases where everolimus was re-administered, the sensitivity of everolimus may have been restored, regardless of resistance to the mTOR inhibitor”appears unclear and does not appear to adequately account for my original criticism. In other word, it would seem quite a nonsense to state that “rechallenge restore sensitivity to everolimus even in the absence of resistance”, if I have not misinterpreted Authors’ words.

Response) Thank you for your valuable comments. We apologize that the description was inappropriate as you indicated. There is no obvious evidence that all tumor cells acquired resistance to everolimus and also no relationship other treatments and restored sensitivity for everolimus. In the duration of other treatments, everolimus sensitive tumor cells may be dominant in tumor mass. So, it is very difficult to describe that “ We speculate that in some cases where everolimus was re-administered, the sensitivity of everolimus may have been restored, regardless of resistance to the mTOR inhibitor”

We have reviewed the description and modified the discussion as follows.

“In this study, there were six cases that were terminated with PD at the time of initial introduction of everolimus, of which 3 had PD, 2 had SD, and 1 had PR as the treatment response to rechallenge with everolimus. In short, half of the patients who was terminated with PD at initial introduction of everolimus were able to disease control at the time of everolimus rechallenge. Therefore, sensitive residual tumor cells for everolimus with initial introduction may be alive and growth for long duration with other treatment options and tumor mass may be shrinkage and achieve stable disease and partial response with everolimus rechallenge. In this regard, since this study was a retrospective analysis of a small number of cases, a detailed analysis that included insights on molecular mechanisms is required in future. Further, studies involving higher number of cases are desirable to derive definitive conclusions on the topic.”

3)    English needs major revision!   It appears that the Authors just reviewed their text, which in several sentences, scattered throughout the paper.

Few examples of this are:

  1. a)    “Everolimus rechallenge may be feasibile and safety Everolimus rechallenge may be a new treatment optionfor patients with advanced PNENs for whom no other treatment option is available”.
  2. b)    “Few treatment options wereavailable for unresectable PNENs. Therefore, physicians sometime must recommend feasible treatment for the advanced PNENs patients with terminated all available treatment and no other treatment options in high volume center.

The sentence is hardly understandable  

  1. c)    “Unfortunately, there is no other clinical trials to overcome everolimus resistance.

Clinical trials do not overcome drug resistance!

Response) Thank you for your valuable feedback. We have proofread the English and revised the text.

  1. a) The description has been corrected as follows (line 37-39, 237-239)

"In conclusion, the outcome and safety of everolimus rechallenge was acceptable treatment. therapy for patients with advanced PNENs for whom no other drug treatment option is available. "

The title has also been changed as follows.

“Clinical outcome of Everolimus Rechallenge in Patients with Pancreatic Neuroendocrine Neoplasms with No Other Treatment Options”

  1. b) The description was deemed unnecessary and deleted.
  2. c) The description was also changed as follows

"Unfortunately, no clinical trials have examined everolimus resistance."